# A New *ABCA3* Gene Mutation c.3445G>A (p.Asp1149Asn) as a Causative Agent of Newborn Lethal Respiratory Distress Syndrome

**DOI:** 10.3390/medicina55070389

**Published:** 2019-07-19

**Authors:** Georgios Mitsiakos, Christos Tsakalidis, Paraskevi Karagianni, Dimitra Gialamprinou, Ilias Chatziioannidis, Ioannis Papoulidis, Ioannis Tsanakas, Vasiliki Soubasi

**Affiliations:** 12nd Neonatology Department and Neonatal Intensive Care Unit, Aristotle University of Thessaloniki, “Papageorgiou” Hospital, Thessaloniki, PC 56403, Greece; 23rd Department of Pediatrics, Aristotle University of Thessaloniki, “Hippokratio” Hospital, Thessaloniki, PC 54642, Greece

**Keywords:** surfactant deficiency, *ABCA3* gene mutation, neonatal respiratory distress syndrome, medical treatment

## Abstract

Mutations in adenosine triphosphate-binding cassette transporter A3 (*ABCA3*) (OMIM: 601615) gene constitute the most frequent genetic cause of severe neonatal respiratory distress syndrome (RDS) and interstitial lung disease (ILD) in children. Interstitial lung disease in children and especially in infants, in contrast to adults, is more likely to appear as a result of developmental deficits or is characterized by genetic aberrations of pulmonary surfactant homeostasis not responding to exogenous surfactant administration. The underlying *ABCA3* gene mutations are commonly thought, regarding null mutations, to determine the clinical course of the disease while there exist mutation types, especially missense variants, whose effects on surfactant proteins are difficult to predict. In addition, clinical and radiological signs overlap with those of surfactant proteins B and C mutations making diagnosis challenging. We demonstrate a case of a one-term newborn male with lethal respiratory failure caused by homozygous missense *ABCA3* gene mutation c.3445G>A (p.Asp1149Asn), which, to our knowledge, was not previously reported as a causative agent of newborn lethal RDS. Therapeutic strategies for patients with *ABCA3* gene mutations are not sufficiently evidence-based. Therefore, the description of the clinical course and treatment of the disease in terms of a likely correlation between genotype and phenotype is crucial for the development of the optimal clinical approach for affected individuals.

## 1. Introduction

Genetic disorders of surfactant metabolism underlie a group of rare diseases with a wide range of clinical manifestations from lethal respiratory distress syndrome (RDS) in term newborns to chronic interstitial lung disease (ILD) in children and adults [1,2]. Adenosine triphosphate-binding cassette transporter A3 (*ABCA3*) gene deficiency is the most frequent genetic cause of RDS and ILD with so far unknown incidence in the term newborn population. ABCA3 is a phospholipid transporter protein critical for surfactant homeostasis, localizes to the membranes of lamellar bodies and is further related to lamellar bodies’ (LBs) maturity; it is a crucial membrane protein involved in the transport of lipids across the biological membranes by hydrolyzing ATP [3]. Bi-allelic null nonsense or frameshift mutations depict the most frequent cause of congenital surfactant deficiency resulting in fulminant severe RDS that is lethal without lung transplantation before the 1st year of life [4,5,6,7]. The “other” mutations including missense, splice site, and in-frame insertion/deletions depict a wide range in presentation, severity, and prognosis from severe neonatal RDS to ILD always regarding modifiers like genetic or environmental factors which may alter disease course [4,5,6]. Mono-allelic mutations represent a partially impaired *ABCA3* transporting function, which is usually presented with mild clinical course [8]. Alternatively, they might act as disease modifiers in lung diseases, such as RDS of prematurity or are associated with mutations in other surfactant-related genes [6,7,8,9,10,11].

Apart from lung transplantation, there are no established therapeutic approaches for inherited disorders of surfactant metabolism [2]; although lung transplantation is the definitive treatment [12], post-lung transplantation morbidities and mortality rate remain considerably high for infants and children with surfactant metabolism genetic disorders [13].

In the present study, we report the clinical course of a Greek patient with a novel homozygous *ABCA3* gene mutation c.3445G>A (p.Asp1149Asn), which, to our knowledge, was not previously reported as a causative agent of newborn lethal RDS. Additionally, we define our differential diagnostic approaches and the application outcomes of the therapeutic strategies.

## 2. Case Report

The patient was a male newborn delivered by spontaneous vaginal delivery at 39 weeks and 2 days after an uncomplicated pregnancy. The birth weight was 3330 g, height and head circumference appropriate for gestational age, and Apgar scores of 8 and 9 at 1 and 5 min, respectively. The patient was the second born of non-consanguineous parents free from ILD family history.

Within 12 h of life, the patient developed signs of respiratory distress which was initially treated with diffused oxygen and gradually deteriorated requiring nasal continuous positive airway pressure and empirical antibiotic therapy. On the third day of life increasing respiratory distress required ventilation with synchronized intermittent positive pressure ventilation. Due to radiological signs of RDS (Figure 1a,b), he received 2 doses of surfactant with only a transient improvement in oxygenation. Echocardiography showed a small atrial septal defect with little signs of pulmonary hypertension and a brief inhaled nitric oxide trial had little effect on the clinical course. With increasing mean airway pressure and FiO_2_, high-frequency oscillatory ventilation (HFOV) was used on the sixth day after birth. At ten post-birth days, he was still ventilated with HFOV (Figure 1c) and showed gradual improvement of oxygenation index from 38 to 23 on the 15th day of life (Figure 2). At this time, ventilation switched to synchronized intermittent positive pressure ventilation, but increasing oxygen need was supported with HFOV. As signs of pulmonary hypertension worsened, he additionally received sildenafil and diuretic treatment. 

Due to the ongoing need for mechanical ventilation, at 2 post-birth weeks, a high-resolution computed tomography (HRCT) of the lungs was performed showing a characteristic pattern of ILD with ground-glass opacities, interstitial emphysema, and pulmonary edema (Figure 3). With suspicion of potential underlying surfactant metabolism genetic disorders, written informed consent from parents was obtained for genetic study. Bronchoscopy and bronchoalveolar lavage (BAL) were not performed due to clinical instability and the parents’ declined consent for an open lung biopsy. Genomic DNA was isolated according to the protocol specifications of Nucleospin Blood kit (Macherey Nagel, Düren, Germany). Next-generation sequencing (NGS) was performed with MiSeq (Illumina, San Diego, CA, USA) in all coding regions and splicing sites of the following genes: *ABCA3* (NM_001089.2), *CSF2RA* (NM_006140.4), *CSF2RB* (NM_000395.2), *SFTPA1* (NM_005411.4), *SFTPB* (NM_198843.2), *SFTPC* (NM_003018.3), and *SFTPD* (NM_003019.4). Amplification of the above genomic regions was carried out by oligonucleotide-based target capture (QXT, Agilent Technologies, Santa Clara, CA, USA). Data analysis was performed using BURROWS-WHEELER ALIGNER (BWA MiSeq Reporter, v2.1.0.0, Palo Alto, CA, USA) and GENOME ANALYSIS TOOLKIT (GATK v1.6-23-gf0210b3) with minimum coverage depth 20X. The c.3445G>A (p.Asp1149Asn) mutation found in the *ABCA3* gene was confirmed by Sanger sequence analysis. This new bi-allelic *ABCA3* mutation has not been described so far in the literature and the databases dbSNP, ExAC, ESP and gnomAD (http://www.ncbi.nlm.nih.gov/SNP, https://www.ensembl.org/index.html, http://evs.gs.washington.edu/EVS, http://gnomad.broadinstitute.org, http://www.genome.med.kyoto-u.ac.jp/SnpDB, http://www.ncbi.nlm.nih.gov/clinvar, http://exac.broadinstitute.org).

At 12 post-birth days, corticosteroids treatment based on protocol treatment for patients with surfactant deficiency caused by mutations in *SP-C* gene (http://www.klinicum.uni-muenchende/childEU/en/hcq4surf_erare/hydroxychloroquine/index.html) was initiated consisting of methylprednisolone pulse therapy (300 mg/m^2^/d intravenously for 3 days with 2-week interval) and oral prednisolone (2 mg/kg/d) received until the patient died with transient improvement in ventilator settings and oxygen need. Treatment with macrolides (azithromycin 10 mg/kg/day for the first day and 5 mg/kg/day for the following 4 days) was applied as an immunomodulator lung function factor. Likewise, surfactant administered 10 times during the clinical course achieved no remarkable improvement of the oxygenation index. Trials for weaning from the HFOV or the high setting in ventilation supporting were just unsuccessful intervals in the insufficient respiratory status of the patient.

He experienced sepsis twice on the 26th and 57th day of life from staphylococcus for which he was treated with broad-spectrum antibiotics and immunoglobulin. He was initially fed with standard formula through a gastric tube which was well tolerated and his weight gain was satisfactory. At 28 post-birth days, he developed temporary cholestasis with 4.0 mg/dl direct bilirubin related to the sepsis event. An elemental formula through the gastric tube was gradually performed until the recovery of clinical presentation and laboratory values. 

At 78 post-birth days of life, the patient was transferred to a pediatric intensive care unit (PICU) and he received treatment with hydroxychloroquine (10 mg/kg/day), corticosteroids (prednisone 2 mg/kg/day), azithromycin (20 mg/kg/day for three days per week), and surfactant (100 mg/kg) without improvement of the clinical status and ventilation. He developed bilateral pneumothorax that required thoracostomy tube drainage. 

The patient remained supported with mechanical ventilation with HFOV and high setting, sedated with fentanyl and midazolam, in muscle relaxation with rocuronium bromide and unable to wean in seriously critical clinical condition until he deceased at post-birth 9 months and 7 days.

## 3. Results 

In this patient report, we describe a novel homozygous *ABCA3* gene missense mutation, the c.3445G>A (p.Asp1149Asn), with lethal early-onset and severe progression, which, to our knowledge, has not previously been described in the literature and the databases dbSNP, ExACH, ESP, gnomAD. The in silico analysis and the patient’s phenotype suggest that this is a pathologic variant and the genetic cause of the disease. According to international best practice, the detection of an identical *ABCA3* bi-allelic gene mutation in another patient with lethal RDS and surfactant deficiency is required to confirm the correlation between the genotype and the relevant clinical syndrome. 

## 4. Discussion and Conclusions

More than 200 *ABCA3* mutations have been reported to date with around three-quarters of patients presented as compound heterozygotes [14]. Regarding the biological consequences of *ABCA3* missense mutations, they are initially considered to be leading to an impaired trafficking or dysfunction of the ABCA3 protein [15]. According to Matsumura et al., *ABCA3* gene mutations may follow two cellular pathways affecting intracellular surfactant processing (type I) and ABCA3 decreased ATP-hydrolysis activity (type II), respectively [16] The complex genetic and molecular pathways underlying congenital surfactant deficiencies pose challenges in designing effective treatment strategies. The opportunities presented by an enhanced understanding of ABCA3 biology for targeted therapeutic strategies have recently been addressed [14]. With a special focus on correlating *ABCA3* gene missense mutations to clinical phenotypes, investigators developed new tools which demonstrate the association between in vitro observations and clinical data [17]. 

Our patient had a new homozygous c.3445G>A (p.Asp1149Asn) gene missense mutation located in exon 23 of the *ABCA3* gene with total (surfactant) deficiency and fatal early onset. Regarding the variant, c.3445G>A (p.Asp1149Asn) in the *ABCA3* gene it is not reported in the literature nor in the following population databases: dbSNP, ESP, ExAC, and gnomAD. However, it is located in a highly conserved residue and bioinformatics analysis suggests it is probably a pathogenic variant. Considering the above and according to ACMG guidelines for the classification of sequence variants [18], we believe that it should be classified as a likely pathogenic variant (class IV). This bi-allelic defect seems to be the genetic cause of this patient’s lethal RDS in conjunction with the ABCA3 deficiency. The in silico analysis showed no other genetic impairment of the *ABCA3* gene or of another relevant surfactant gene. Recent studies by Schindlbeck et al. [17] in terms of depicting correlations between missense mutations in *ABCA3* gene and the clinical severity of the disease by investigation of molecular pathomechanisms added new data in the bibliography. For instance, a full-term newborn with the homozygous combination of p.Q215K and p.R288K missense *ABCA3* gene mutations was presented with severe respiratory distress syndrome and early neonatal death due to ABCA3 malfunction [19,20]. The low impact of the variant p.R288K mediated by the variant p.Q215K results in a severe phenotype. The studied findings support the concept of the variants’ interaction. Thus, it is not easy to provide a direct prediction of their clinical impact of the missense mutations, which may vary from milder to severe phenotype imposing investigation in biological aspects in order to be classified. The life expectancy of our patient was expected for the majority of patients with surfactant deficiency due to biallelic missense mutations in *ABCA3* genes.

Clinical and radiological signs of patients with *ABCA3* gene mutations overlap with those encountered in other genetic disorders of surfactant metabolism. The diagnostic approach is mainly based on genetic studies and confirmed if applicable by lung biopsy [21]; the diagnostic approaches include genetic testing, echocardiography, high-resolution CT, bronchoscopy when appropriate and the definitive investigation of lung biopsy to establish a precise diagnosis [22]. Electron microscopy should be performed systematically, since it may disclose alterations in LBs, which are highly specific and distinct for *ABCA3* mutations from those observed in SP-C, SP-B, and TTF deficiencies [19]. We defined our diagnosis with DNA sequencing. The parents declined the lung biopsy so imaging by LB’s morphology electron microscopy is not available. In addition, we did not perform parental DNA sequencing which could distinguish whether this was a recessive inheritance of the mutation or a uniparental disomy or a de novo mutation and this is a potential limitation in our study. HRCT and BAL provide information for the underlying ILD and the severity of lung inflammation, respectively, but their role is equally supporting and not critical to confirm the genetic diagnosis. The HRCT of our patient revealed a damaging lung pattern consistent to ILD while BAL was not performed.

Therapeutic protocols with corticosteroids, macrolides and hydroxychloroquine in combination are suggested for patients with interstitial lung disease by SP-C deficiency and are empirically used in patients with *ABCA3* gene mutations. Regarding reports of constant clinical improvement in SFTPC [23] and elective improvement in specific *ABCA3* mutations by hydroxychloroquine administration, this medical management may play a role in modifying the clinical course of the disease [24,25]. Some patients respond with improvement to chronic interstitial lung disease while others never achieve independence from mechanical ventilation and, therefore, require lung transplantation. Ciantelli et al. described a newborn with a severe RDS by a compound heterozygous mutation of the *ABCA3* gene of two new variants in whom persistent surfactant administration and corticosteroids therapy led to no improvement in the clinical course [26]. In contrast, Tan et al. reported a case of ABCA3 lung disease that demonstrated improvement after systemic glucocorticosteroid administration [27]. Winter et al., report a patient with a homozygous *ABCA3* gene mutation in whom neither the application of corticosteroids, hydroxychloroquine, and macrolides nor any mechanical ventilation mode had a positive effect on the clinical course [28]. Kroner et al. [20], retrospectively analyzed baseline and outcome characteristics of 40 patients with two disease-causing *ABCA3* mutations collected between 2001 and 2015. The investigators showed that treatment with an exogenous surfactant, systemic steroids, hydroxychloroquine, and whole lung lavages had apparent but many times transient effects in individuals [20]. Our patient had a lethal early-onset disease with severe progression in which long-term respiratory support and supplemental oxygen, recurrent surfactant administration, macrolides, corticosteroids, and hydroxychloroquine achieved no remarkable clinical improvement.

Moreover, although the mentioned lung transplantation has a 5-year survival of 50%, the donor organ shortage and the limited number of transplantation centers should not be ignored. For these reasons, our patient was not included in the transplantation list [26,28] of the available expertise institutes while it is worth mentioning that in Greece there is not an established pediatric lung transplantation center.

The present patient report refers to term newborns presented with respiratory distress syndrome with no obvious cause and rapid progression to lethal respiratory failure soon after birth that is not responsive in any mode of ventilation or drug intervention. This clinical scenario requires investigation of genetic disorders of surfactant metabolism. The report of new *ABCA3* gene mutations, the description of clinical course and response to management contributes to the improved knowledge of the disease, the evaluation of the clinical course and the so far insufficient therapeutic interventions in diseases of a quite rare incidence but of a wide genetic and phenotypic diversity.

## Figures and Tables

**Figure 1 medicina-55-00389-f001:**
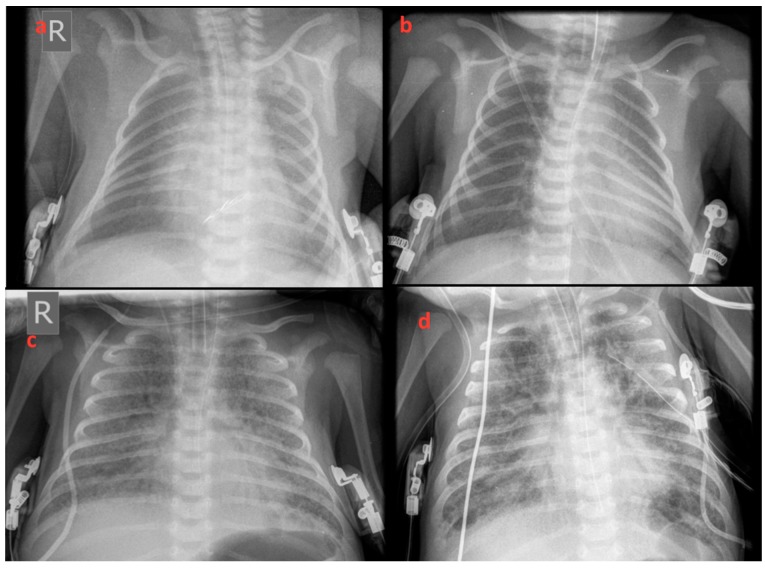
X-rays revealed increased whitening of the lungs consistent with respiratory distress syndrome (RDS) progressively deteriorating from the 1st to 75th day of life. (**a**) 1st, (**b**) 3rd, (**c**) 15th, and (**d**) 75th day of life.

**Figure 2 medicina-55-00389-f002:**
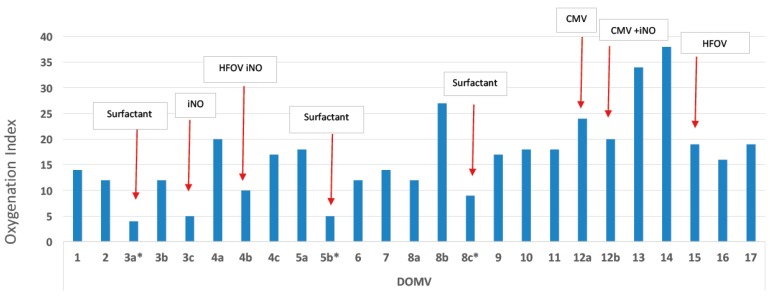
Variance of the oxygenation index during the early 17 days of life, while the patient was treated with mechanical ventilation. The red arrows indicate oxygenation index after treatments (the intervention resulted in lowering the oxygenation index). iNO: inhaled nitric oxide, HFOV: high frequency oscillatory ventilation, CMV: conventional mechanical ventilation, DOMV: day of life on mechanical ventilation.

**Figure 3 medicina-55-00389-f003:**
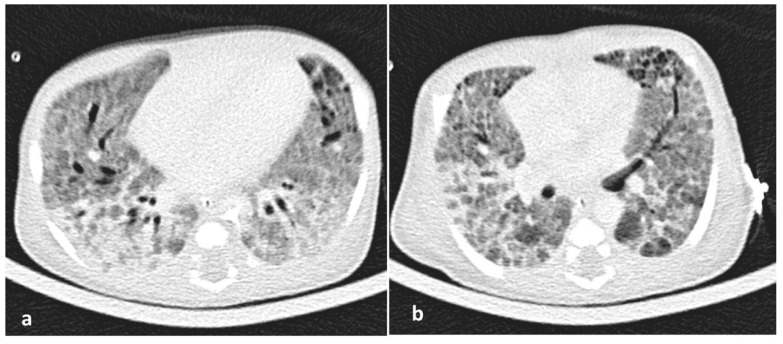
(**a**,**b**). A high-resolution CT (HRCT) of the lungs showing patchy areas of ground glass attenuation with thickening of interlobular septae also called ‘crazy paving’ pattern at 15th day of life.

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
