# Peer review of "A New ABCA3 Gene Mutation c.3445G>A (p.Asp1149Asn) as a Causative Agent of Newborn Lethal Respiratory Distress Syndrome"

_medicina, 2019, doi:10.3390/medicina55070389_

Round 1
Reviewer 1 Report
This is a revised manuscript of a Greek infant with fatal case of a novel ABCA3 homozygous mutation. The clinical course and treatment of the infant is described in detail, and although the content itself lack significant originality, the detailed clinical vignette may be of interest to readers, especially neonataologists, pulmonologists, and general pediatricians.
1. Figures and figure captions are not available in this manuscript.
2. “c3445G>A(p-ASP1149Asn)” This is not a correct expression of gene mutation. Please refer to standard nomenclature of gene mutations.
3. Listing the various previous cases of compound missense mutation of ABCA3 located far from the proband’s mutation does not support the pathogenicity of the newly found mutation. Although functional study was not done, do the authors have any hypothesis of why the found mutation is pathogenic? Maybe regarding the domain or function of the mutation site?
4. The discussion is overall redundant and lengthy, especially regarding the different consequences of “null” and “other” mutations.
5. Overall, the manuscript would benefit substantially from English editing. There are many grammatical errors and sentences with loose structures. Also, software or company names should be capitalized, and for softwares, versions should be mentioned.
Author Response
1. Figures and figure captions are not available in this manuscript.
Figures and the relative captions are now available, please find them in the supplementary file which has been attached to the draft.
2. “c3445G>A(p-ASP1149Asn)” This is not a correct expression of gene mutation. Please refer to standard nomenclature of gene mutations.
According to the HGVS standard nomenclature is c.3445G>A (p.Asp1149Asn). Thank you for the comment we have already checked it to the text.
3. Listing the various previous cases of compound missense mutation of ABCA3 located far from the proband’s mutation does not support the pathogenicity of the newly found mutation. Although functional study was not done, do the authors have any hypothesis of why the found mutation is pathogenic? Maybe regarding the domain or function of the mutation site?
Regarding the variant c.3445G>A (p.Asp1149Asn) in ABCA3 gene I would like to inform you that it is not reported in the literature nor in the following population databases: dbSNP, ESP, ExAC and gnomAD. However, it is located in a highly conserved residue and bioinformatics analysis suggests it is probably a pathogenic variant. Considering the above and according to ACMG guidelines for the classification of sequence variants (Standards and Guidelines for the Interpretation of Sequence Variants: A Joint Consensus Recommendation of the American College of Medical Genetics and Genomics and the Association for Molecular Pathology, Genet Med. 2015 May ; 17(5): 405–424.) we believe that it should be classified as likely pathogenic variant (class IV)
4. The discussion is overall redundant and lengthy, especially regarding the different consequences of “null” and “other” mutations
Thank you help us with your comments improve the manuscript. We tried to condense and reorganize the sections mostly by avoiding overlaps.
5. Overall, the manuscript would benefit substantially from English editing. There are many
grammatical errors and sentences with loose structures. Also, software or company names should be capitalized, and for softwares, versions should be mentioned. Grammatical and structural errors have been corrected. Company names and software have been capitalized.
Reviewer 2 Report
1. Introduction, page 3, line 45. The incidence of ABCA3 deficiency is not known. The manuscript Wambach JA et al Pediatrics, 2012 estimates a disease frequency, but the prevalence of predicted deleterious variants in large databases (gnomad) likely overestimates the disease frequency. Additionally, the majority of variants are private, it is difficult to know which are truly pathogenic. The authors should clarify.
2. Intro, page 3, line 47. Unclear what is meant by "further related to lamellar bodies' mature morphology. Please clarify.
3. Intro, page 3, line 52. Seems a word is missing: "...insertions/deletions the consequences cannot be easily."
4. Therapeutic strategies including steroids, macrolides, hydroxychloroquine have also been tried for patients with ABCA3 mutations (Kroner et al, Thorax, 2017), not just SPB deficiency. Recommend including this reference.
5. Case report, page 5. Was clinical or research sequencing obtained for this patient?
6. Case report, page 6- authors mention oral prednisolone was continued until parent 'was discharged'- perhaps it would be better to say prednisolone was continued until death.
7. Case report, Page 7- why was patient treated with immunoglobulin? Was an immunodeficiency suspected or is this routine treatment with clinical sepsis?
8. Discussion, Page 10- missense variants have been shown to disrupt intracellular trafficking of ABCA3 to the lamellar bodies- the statement that missense variants are "likely low impact" is not accurate. Their patient was homozygous for a missense variant and had lethal RDS.
9. There is much overlap in the material presented in the introduction and discussion and this could be condensed.
10. The discussion could benefit from re-organization.
Author Response
1. Introduction, page 3, line 45. The incidence of ABCA3 deficiency is not known. The manuscript Wambach JA et al Pediatrics, 2012 estimates a disease frequency, but the prevalence of predicted deleterious variants in large databases (gnomad) likely overestimates the disease frequency. Additionally, the majority of variants are private, it is difficult to know which are truly pathogenic. The authors should clarify.
The incidence has been estimated approximately for European and African population in 3.6% by M.F, Beers and S.Mulgeta but the overall prevalence is difficult to accurately be estimated. Regarding this inconvenience we excluded this from the manuscript.
2. Intro, page 3, line 47. Unclear what is meant by "further related to lamellar bodies' mature morphology. Please clarify.
It refers to how ABCA3 affects the biogenesis of LBs.
3. Therapeutic strategies including steroids, macrolides, hydroxychloroquine have also been tried for patients with ABCA3 mutations (Kroner et al, Thorax, 2017), not just SPB deficiency. Recommend including this reference.
This reference has already been included.
4. Case report, page 5. Was clinical or research sequencing obtained for this patient?
We have already answered this question in previously revision. It was obtained in a research laboratory specially for this patient.
5. Case report, page 6- authors mention oral prednisolone was continued until parent 'was discharged'- perhaps it would be better to say prednisolone was continued until death
It has been corrected in the draft.
6. Case report, Page 7- why was patient treated with immunoglobulin? Was an immunodeficiency suspected or is this routine treatment with clinical sepsis?
The patient was immunodeficient due to sepsis and furthermore due to prednisolone long-term course. Thus was treated supportively with immunoglobulin.
7. Discussion, Page 10- missense variants have been shown to disrupt intracellular trafficking of ABCA3 to the lamellar bodies- the statement that missense variants are "likely low impact" is not accurate. Their patient was homozygous for a missense variant and had lethal RDS
We excluded this from the draft.
9-10. There is much overlap in the material presented in the introduction and discussion and this could be condensed. The discussion could benefit from re-organization.
Thank you help us with your comments improve the manuscript. We tried to condense and reorganize the sections mostly by avoiding overlaps.
Reviewer 3 Report
The authors have answered to almost all my queries
I would not use demonstrate in the abstract in absence of fonctionnal analysis.
As far as we know, patients with monoallelic ABCA3 mutation do not present symptoms.
The genes are in italic.
ABCA3 does not help AEC2 to produce surfactant.
The discussion is too long.
Author Response
The authors have answered to almost all my queries
The authors thank the reviewer for the revision of the manuscript. As we previously mentioned we tried to improve the quality of the draft by condensing and reorganizing the sections.
Round 2
Reviewer 1 Report
The authors have generally answered adequately to the comments.
Author Response
The authors thank the reviewer for the critical appraisal of the draft.
Regarding the current bibliography we overall believe that a well-structured scientific approach was achieved and the outcomes are supported equally adequate.
You will find attached in the revised manuscript the versions of the used softwares as you have recommended to include them in a previous revision.
Reviewer 2 Report
The authors have made substantial improvements to the manuscript. Recommend a few minor edits:
Intro, page 3, line 56: References for monoallelic ABCA3 variants are missing. Recommend adding Wambach JA et al Pediatr, 2012; Wittmann T Mol Med 2016, Naderi HM Am J Med Genet 2014.
Discussion, page 10, lines 182 and 197: the patient presented is homozygous for a missense ABCA3 mutation, rather than nonsense or frameshift mutations. Thus the term "complete ABCA3 deficiency" is misleading as the authors do not present staining for ABCA3 in the lung tissue nor functional data regarding this mutation.
Discussion, page 10: Recommend removing the sentence: "It is important to note that this new homozygous missense mutation of our own patient cannot be accurately classified yet. It is not a stop codon or a frameshift mutations as the most other defects that have been identified so far for patients with fatal progression of the disease." This information has already been addressed in other sections of the manuscript and can be removed for brevity.
Discussion, page 10, lines 188-189: Unclear what is meant by "clinical severity of the disease, added new data in bibliography."
Discussion, page 10, lines 198-199- the sentence "The identification of the same gene defect in another patient seems to be a prerequisitie for the molecular classification of this mutation and for further phenotype correlation" can be removed as similar information is included in other sections of the text.
Discussion, page 11, lines 204-205: some may argue that lung biopsy histology is less precise/ definitive than genetic sequencing for genetic disorders of surfactant metabolism as there is considerable histologic overlap.
Author Response
The authors have made substantial improvements to the manuscript.
On behalf of all the authors honestly we appreciate your constructive contribution in the improvement of the manuscript.
Recommend a few minor edits:
Intro, page 3, line 56: References for monoallelic ABCA3 variants are missing. Recommend adding Wambach JA et al Pediatr, 2012; Wittmann T Mol Med 2016, Naderi HM Am J Med Genet 2014.
As the references were missing we added those that were recommended.
Discussion, page 10, lines 182 and 197: the patient presented is homozygous for a missense ABCA3 mutation, rather than nonsense or frameshift mutations. Thus the term "complete ABCA3 deficiency" is misleading as the authors do not present staining for ABCA3 in the lung tissue nor functional data regarding this mutation.
The term “complete” removed regarding this inconvenience.
Discussion, page 10: Recommend removing the sentence: "It is important to note that this new homozygous missense mutation of our own patient cannot be accurately classified yet. It is not a stop codon or a frameshift mutations as the most other defects that have been identified so far for patients with fatal progression of the disease." This information has already been addressed in other sections of the manuscript and can be removed for brevity.
The sentence was removed.
Discussion, page 10, lines 188-189: Unclear what is meant by "clinical severity of the disease, added new data in bibliography."
We mean that the outcomes of the authors survey in this field added substantial information in the bibliography. We have already corrected it in the text.
Discussion, page 10, lines 198-199- the sentence "The identification of the same gene defect in another patient seems to be a prerequisitie for the molecular classification of this mutation and for further phenotype correlation" can be removed as similar information is included in other sections of the text.
We have removed it as it was overlapped.
Discussion, page 11, lines 204-205: some may argue that lung biopsy histology is less precise/ definitive than genetic sequencing for genetic disorders of surfactant metabolism as there is considerable histologic overlap.
Taking in account the bibliography, the histologic findings in SP-C, SP-B, ABCA3 mutations provide similar distinctive- usually overlapped- features which further determine histologically the interstitial lung disease. That is why we perform DNA sequencing. The morphology of LBs in ABCA3 mutations maybe suggests an additional information in the diagnosis by using lung biopsy histology.
This manuscript is a resubmission of an earlier submission. The following is a list of the peer review reports and author responses from that submission.
Round 1
Reviewer 1 Report
This is a very detailed and clear clinical vignette of a term baby with lethal RDS, diagnosed with a novel ABCA3 mutation. Here I attach my opinion on this manuscript. Please refer to it.

Author Response
Please check the file attached

Reviewer 2 Report
In "A new ABCA3 gene mutation c3445G>A (p-ASP1149Asn) as a causative agent of newborn lethal respiratory distress syndrome" the authors present an infant with neonatal respiratory failure and a novel ABCA3 variant p.Asp1149Asn.
Major comment: While the specific ABCA3 variant is novel the clinical course and radiographic findings are similar to other infants with ABCA3 deficiency. Manuscript would be strengthened by addition of results from parental samples- is this recessive inheritance or uniparental disomy which has been reported for ABCA3 deficiency.
Minor comments
Abstract
1. There are causes of ILD other than "genetic aberrations of pulmonary surfactant homeostasis not responding to exogenous surfactant admninistration" and this statement should be clarified.
2. While biallelic nonsense or frameshift variants are associated with neonatal presentation and death before 1 year of age without lung transplantation, the presentation and course of infants with other types of mutations is difficult to predict. The authors statement that "the underlying mutations determine the clinical course of the disease" is likely an oversimplification as other genetic or environmental factors likely influence disease course.
Introduction
1. The statement "ABCA3 transports compounds across the biological membranes" should be more specific- ABCA3 transports phospholipids across the membrane of malellar bodies.
2. Unclear what is meant by the statement that "benign variants are frequently subclinical"- this should be clarified.
3. Infants with genetic disorders of surfactant metabolism typically have a transient response to exogenous surfactant (as their patient did) rather than no response.
4. The introduction is somewhat repetitive and canc be shortened/ condensed.
Case report
1. Was the genetic testing performed on a research or clinical basis?
2. Unclear why SFTPA1 and SFTPD sequencing was performed as genetic encoded disruption of these genes have not been associated with neonatal respiratory failure.
3. Authors should provide websites for databases (dbSNP, etc). Authors should consider including gnomad results as well.
4. Authors should further describe what is meant by "transient improvement" in repsonse to methylprednisolone and prednisolone- why were both steroids used or were these medications used successively and for how long?
5. Perhaps the term 'standard' infant formula is more appropriate than 'normal'.
6. Authors should consider further description of cholestasis as this is not a common finding for infants with ABCA3 deficiency- what was the level of direct bilirubin? Was this temporally related to the sepsis events?
7. How were the authors able to determine that that pneumothorax occurred at multiple sites?
8. What sedation medications were used? Did the infant require muscle relaxation?
Results
1. Authors should state which in silico algorithms were used to predict variant pathogenicity.
2. Authors should clarify the statement: "SPB and SPC are the 2 hydrophobic proteins which are assembled into intracellular organelles" as this is not accurate.
3. Much of the information in the introduction and discussion is redundant and could be better organized and condensed.
4. Authors mention that ABCA3 is expressed in other tissues- any other non-lung phenotypes associated with biallelic variants in this gene?
5. References for cataract-microcornea and leukemia associations are missing.
6. The genotype-phenotype correlation for ABCA3 variants (type I vs. type II) is not well characterized and should be clarified.
7. Some nonsense or frameshift variants are likely to result in nonsense mediated decay rather than be type I mutations.
8. The life expectancy of the patient is probably longer than predicated for patients with biallelic nonsense/ frameshift variants, but similar to other patient with biallelic missense variants.
9. Unclear what is meant by the sentence "The recognition of another similar gene defect is a prerequisite for the classification of this mutation" and should be clarified.
10. Some may argue that there is also overlapping histologic findings for patients with biallelic ABCA3 variants and thus lung biopsy may not provide a precise diagnosis.
11. Unclear what is meant by "a wide DNA sequencing" and should be clarified.
12. The recessive variant is unlikely to be de novo and should be clarified in the text. Parental samples would help to determine if the variant is inherited in a recessive manner or as the result of uniparental disomy.
13. The Kroner C et al Thorax 2017 article has additional information about treatment and should be included as a reference.
14. While the clinical scenario of a term infant with respiratory distress syndrome without obvious cause and rapid progression to lethal respiratory failure is rare, ABCA3 deficiency should be considered in these patients and likely explains a significant proportion of these infants. The first sentence of the last paragraph should be clarified.
Graph 1. It is unclear whether the treatments indicated with the red arrows were before
or after the OI.
Figure 1. Please indicate the age of the patient for the different images. Also, since the patient lived for several months, would be interesting to include chest radiographs from later in the disease course (rather than 4 images from the first 3 weeks of life)
Figure 2. Are all images from the same chest CT? If so, perhaps focusing on 1-2 images from this study would be sufficient. Consider including arrows to indicate notable findings.
Figure 3. This figure is not necessary and can be removed.
General comments
1. The manuscript should be edited for grammar and consistency of genetic nomenclature (e.g., c.3445G>A; p.Asp1149Asn).
2. The references are not correctly cited and some key references are missing.
Author Response
Please check the file attached

Reviewer 3 Report
The author report an interesting case of neonatal RDS associated wiht bi allelic missense ABCA3 mutation.
The astract and the introduction are misleading, and the bibliography is inadequate. Please, make sure you cite up-to-date bibliography
ABCA3 mutations has been previsouly associated with neonatal RDS. By the way missense ABCA3 mutation has been associated wiht midler phenotype compared to null mutation
Some treatmnet have been associated wiht clinical improvement
The genetic conclusion does not report the prevalence of this variant in the general population and does not fullfill the ACMG guidelines.
Author Response
Please check the file attached

Round 2
Reviewer 1 Report
The authors have responded to the reviewer’s comments and made substantial changes to the manuscript in accordance with the suggestions.
Minor comments:
1. The ABCA3 gene needs to be distinguished from ABCA3 protein by italicizing. The current manuscript many times does not italicize the gene ABCA3. Please review thoroughly, including the abstract, and italicize if it is used as a gene name.
2. It needs to be “Sanger”, not “SANGER”. (Line 108)
Reviewer 2 Report
In "A new ABCA3 gene mutation c3445G>A (p-ASP1149Asn) as a causative agent of newborn lethal respiratory distress syndrome" the authors present an infant with neonatal respiratory failure and a novel homozygous ABCA3 variant p.Asp1149Asn.
Major comment: While the specific ABCA3 variant is novel, the clinical course and radiographic findings are similar to other infants with ABCA3 deficiency. As this variant is novel and they do not present histologic or functional data, the authors should include the results from specific in silico prediction programs and cite these references. The introduction and the discussion are very lengthy for the information presented and would from condensing and reorganizing. The authors repeatedly mention that the underlying mutations determine the clinical course of the disease. While this is well described for biallelic nonsense or frameshift variants which are associated with neonatal presentation and death before 1 year of age without lung transplantation, the genotype-phenotype correlation for other variants is less clear. Likely other genetic or environmental factors likely influence disease course.
Minor comments
Abstract
1. Unclear what is meant by “as a result of developmental deficits”- does this refer to preterm infant or patients with ILD?
2. The sentence “The underlying ABCA3 gene mutations are commonly thought to determine the clinical course of the disease while exist mutation types whose effects on surfactant proteins are difficult to predict” could be clarified.
3. Abstract, lines 26-27. The new data presented in the manuscript does not specifically demonstrate genotype and phenotype correlation. While each case helps to correlated genotype-phenotype correlation, there are not enough patients presented here to draw firm conclusions.
4. Authors mention development of specific therapies in abstract- might be better placed in the discussion as their data do not suggest new information about therapies.
Introduction
1. Page 1, line 37- authors mention incidence of ABCA3 deficiency is unknown but then reference frequency of ABCA3 mutations in 2 ethnic groups in the discussion.
2. Page 1- the statement “The specific underlying mutations of ABCA3 gene determine the highly variable severity and clinical course of the disease.” While this is true for null/null genotypes, the factors that influence disease severity/ progression for other types of mutations are not known (e.g., specific mutation, environmental, other modifiers).
3. Page 2, line 44-45. Unclear what is meant by “Mono-allelic defects and variants”- authors should clarify.
4. Page 2, lines 47-48: The statement: “Mono-allelic missense ABCA3 damaging and benign variants are frequently subclinical in terms of common synonymous variants which did not increase neonatal RDS risk” needs to be clarified.
Case report
1. Were the DNA samples obtained on a genetic or research basis? Was the sequencing performed in a research or clinical laboratory?
2. The authors reference dbSNP, ExAC, ESP, and gnomAD- recommend adding these references and/or websites to the text.
3. Authors should cite specific reference for the “protocol treatment for patients with surfactant deficiency caused by mutations in SP-C gene”.
4. With which Staphylococcal species was he infected? Was this related to presence of a central line?
Results
1. Which in silico analyses were used to interpret functionality of the ABCA3 variants? Authors should provide specific citations.
2. Term recommended by American College of Medical Genetics is “pathogenic variant” as opposed to “pathological mutation”.
Discussion
1. Page 6, lines 192-193- some missense variants present with severe neonatal respiratory failure- the term “low impact” is misleading and does correctly describe the range of phenotypes associated with missense variants in ABCA3.
2. Page 6, lines 198-199. While it is speculated that the specific ABCA3 mutations directly determine phenotype, this correlation has not been well described for mutations other than nonsense and frameshift. Other factors (genetic, environmental, etc) may alter disease course.
3. Page 6, lines 206-208. Not all compound heterozygous type I/type II mutations result in lethal RDS- this statement should be clarified.
4. Monoallelic variants in ABCA3 typically result in reversible RDS. However, there have been some cases of patients with single ABCA3 mutations and phenotypes of lethal RDS or interstitial lung disease.
5. Page 7, lines 262-264. Information about lack of parental samples might fit better in the results section.
6. Page 7, 278-281. Authors mention inclusion criteria of lung transplantation are very strict, but do not indicate these specific factors. Recommend clarifying this statement.
Figure 2. The arrows indicate interventions, but the timing is unclear. Are the arrows indicating interventions done in response to the low OI or did the intervention result in lowering the OI. Consider placing arrows before or after bars as relevant.
Figure 3. The authors present 4 images from the HRCT- consider pointing out specific findings or just including 1-2 larger images.
Reviewer 3 Report
The authors answered to my questions